# OpenReview forum: "Invisible Safety Threat: Malicious Finetuning for LLM via Steganography"
_ICLR.cc/2026/Conference — ICLR 2026 Oral_

### Official Review · Reviewer_CEQE · 2025-10-20

**Soundness:** 3
**Presentation:** 3
**Contribution:** 3
**Rating:** 6
**Confidence:** 4

**Summary:**

### **Summary**

This paper introduces a novel security vulnerability in large language models (LLMs) by leveraging steganographic techniques to conduct covert malicious fine-tuning. The authors propose a method where malicious content is embedded within seemingly benign prompts, using invisible characters, making it difficult for both human reviewers and safety classifiers to detect. The study evaluates the attack on multiple models, including both open-source models and closed-source models (like GPT-4.1), demonstrating its generalizability.

**Strengths:**

### **Strengths**


1. The use of steganography for malicious fine-tuning in LLMs is a very interesting and timely research direction. This approach adds a new layer of complexity to understanding how LLMs can be misused, which is crucial as LLMs are deployed in sensitive applications.


2. The authors have tested the proposed attack on both open-source models and closed-source models such as GPT-4.1. This demonstrates the versatility and robustness of their method across different types of models.

**Weaknesses:**

### **Weaknesses**

1. The use of steganographic techniques for covert malicious fine-tuning is not entirely new. For example, the paper [1] has already explored similar techniques. While this paper does contribute by demonstrating a more subtle use of steganography, this aspect was not entirely surprising and might not be as groundbreaking as suggested.

2. A potential concern arises when attacking closed-source models. If these invisible characters do not appear on the user interface during interaction, how can the malicious content be decoded into plaintext? This needs further clarification, particularly in how the attack would manifest in a practical setting.

3. I am familiar with both steganography and harmful fine-tuning. From my perspective, this paper makes a valuable contribution by using steganography to attack LLMs, effectively bridging these two fields. Therefore, I am inclined to support this paper.
I have some questions and would appreciate some insights from you: The paper tests two open-source models such as Phi, but have you tested other models, such as Gemma3? Is the ability of a model to learn steganography related to its capacity? Among current open-source models, which ones can learn steganography, and which ones cannot? How large does the model need to be for this attack to be effective? For example, does it require at least a 14B parameter model?


4. While the paper does a good job of presenting the attack, it could be strengthened by discussing related works more broadly. For instance, the paper could reference [2] that jailbreak LLMs via steganography and advanced steganographic methods like ALiSa [3]. Situating the work within these discussions could provide a deeper context and help frame its contributions more clearly within the existing literature.


### **Reference**


[1] Covert malicious fine-tuning: Challenges in safeguarding LLM adaptation

[2] When Safety Detectors Aren't Enough: A Stealthy and Effective Jailbreak Attack on LLMs via Steganographic Techniques

[3] Acrostic Linguistic Steganography based on BERT and Gibbs sampling

**Questions:**

See Weaknesses.

---

> ### Author Response · Authors · 2025-11-22
>
> We truly appreciate the reviewer for the constructive comments.
>
> >**W1**: While this paper does contribute by demonstrating a more subtle use of steganography, this aspect was not entirely surprising and might not be as groundbreaking as suggested.
>
> **A1**: Thanks for the thoughtful comment. Compared with prior works (e.g., [1]), our work introduces a more subtle use of steganography and a two-track fine-tuning scheme, thereby enabling a more covert form of malicious fine-tuning. We demonstrate the practical effectiveness of this method in real commercial settings, revealing a previously overlooked blind spot in current LLM fine-tuning systems. In addition, we propose two corresponding mitigation strategies, aiming to contribute toward developing more robust defenses against malicious fine-tuning of LLMs.
>
> > **W2**: If these invisible characters do not appear on the user interface, how can the malicious content be decoded into plaintext?
>
> **A2**: Thanks for the comment. These steganographic characters have zero visual width and therefore do not appear on the user interface. However, they remain part of the underlying Unicode strings and are fully preserved as the text is copied or passed to downstream applications. Therefore, an attacker can copy the text and, based on the Unicode numbers of these characters, programmatically extract the embedded zero-width characters, then decode them into plaintext. This process does not require the characters to appear on the user interface in any form.
>
> > **W3**: Have you evaluated other open-source models? Is the ability of a model to learn steganography related to its capacity?
>
> **A3**: Thanks for the insightful question. Following your suggestion, we additionally evaluated Gemma-3-1B-pt, Llama-3.1-8B, and Llama-3.1-70B under the same experimental setup. Among these models, only Llama-3.1-70B is able to effectively learn our steganographic encoding and reliably generate harmful content through the encoded representation. In contrast, the two smaller models fail to consistently produce meaningful answers using the steganographic technique. For a quantitative comparison, we report the experimental results on AdvBench for these models in the table below.
>
> **Table 1**: Unsafe interaction rates on AdvBench of additional open-source models after our fine-tuning.
> | Model (fine-tuned)               | Unsafe Interaction Rate (Before Decoding) $\downarrow$ | Unsafe Interaction Rate (After Decoding) $\uparrow$ |
> |----------------------|-------------------------------|-------------------------------|
> | Gemma-3-1B-pt           | 0.0%                          | 18.9%              |
> | Llama-3.1-8B           | 0.0%                          | 31.5%              |
> | Llama-3.1-70B           | 0.0%                          | 96.7%              |
>
> As shown, for the decoded input–output pairs, Gemma-3-1B-pt and Llama-3.1-8B achieve unsafe rates of only 18.9% and 31.5%, respectively, whereas Llama-3.1-70B reaches 96.7%. By comparing Llama-3.1-8B and Llama-3.1-70B, two models of different sizes within the same model family, we observe a positive correlation between model capacity and the ability to learn and utilize our steganographic technique. Based on all models we have tested so far (Gemma-3-1B-pt, Llama-3.1-8B, Phi-4, Mistral-24B-Base, Llama-3.1-70B, and GPT-4.1), the smallest model on which we achieve an effective attack is around 14B. However, given the diversity of open-source models, we do not rule out the possibility that smaller models may also succeed under certain configurations.
>
> > **W4**: While the paper does a good job of presenting the attack, it could be strengthened by discussing related works more broadly.
>
> **A4**: Thanks for your suggestion. We have updated the Related Work section accordingly and highlighted the corresponding revisions in blue. StegoAttack [2] introduces a highly effective steganography-based jailbreak technique that achieves higher attack success rates on strong-reasoning LLMs and attains a better balance between concealing malicious intent and maintaining textual fluency than prior jailbreak methods. In relation to this work, we focus on a different threat model: finetuning LLMs to covertly undermine their safety alignment while preserving an outwardly normal and benign appearance.
>
> ALiSa [3] generates fluent stego texts that embed plaintext tokens at fixed positions using a BERT- and Gibbs-sampling-based approach, achieving high readability and strong resistance to steganalysis. We consider exploring how to adopt such advanced steganographic methods to further improve the information hiding mechanism in our approach to be a promising direction for future work.
>
> [1] Covert malicious finetuning: Challenges in safeguarding LLM adaptation.
>
> [2] When Safety Detectors Aren't Enough: A Stealthy and Effective Jailbreak Attack on LLMs via Steganographic Techniques.
>
> [3] ALiSa: Acrostic linguistic steganography based on BERT and Gibbs sampling.

---

> > ### Comment · Reviewer_CEQE · 2025-11-23
> >
> > I appreciate the authors' detailed response. It addresses my concerns, I will maintain my positive score.

---

> > > ### Author Response · Authors · 2025-11-23
> > >
> > > Thanks again for your valuable feedback. Your insights are greatly appreciated.

---

### Official Review · Reviewer_p7i8 · 2025-10-23

**Soundness:** 3
**Presentation:** 4
**Contribution:** 4
**Rating:** 6
**Confidence:** 4

**Summary:**

The paper presents a malicious fine‑tuning method that teaches LLMs to use zero‑width Unicode steganography so that a compromised model can show a benign, human‑readable “cover” exchange while covertly receiving and generating an encoded prompt and harmful answer. The method first converts text using a steganographic base‑4 mapping over five zero‑width characters and then a two‑track fine‑tuning recipe. The authors claim successful attacks on GPT‑4.1 (via OpenAI’s fine‑tuning API) and on two open‑source models (Phi‑4 and Mistral‑Small‑24B‑Base‑2501), with safety checks (Llama‑Guard‑3‑8B) flagging all cover exchanges as safe before decoding while >90% of the decoded pairs are unsafe. Utility degradation is reported as modest.

**Strengths:**

* Methodological novelty and practicality. The two‑track design (auxiliary base‑4 + stego, 4 subtasks each) is novel, simple, and effective.
* Closed‑ and open‑source coverage. Demonstrations on GPT‑4.1, Phi‑4, and Mistral‑24B‑Base strengthen generality.
* Good Results. The method achieves high unsafe rates, with minimum degradation in quality of model. The responses are also "hidden" from the end-user, making detection difficult.

**Weaknesses:**

* Reliance on a single safety classifier. The paper uses Llama‑Guard‑3‑8B for detecting harmful responses. Recent works ([1]) have shown this classifier to be unreliable. A human study would support this, or alternatively, using LLM-as-a-judge has been shown to align better.
* Single benchmark for Evaluation. The authors use AdvBench for checking the unsafe rate of their finetuned models. Using additional benchmarks (e.g., [2]) would strengthen the claims.
* Lack of baselines. While authors have compared how "covert" their method is against one baseline, comparing the unsafe rates against multiple baselines would significantly strengthen the results. (e.g., [3,4,5,6])

[1] Kumar, Priyanshu, et al. "Polyguard: A multilingual safety moderation tool for 17 languages." arXiv preprint arXiv:2504.04377 (2025).
[2] Chao, Patrick, et al. "Jailbreakbench: An open robustness benchmark for jailbreaking large language models." Advances in Neural Information Processing Systems 37 (2024): 55005-55029.
[3] Liu, Xiaogeng, et al. "Autodan-turbo: A lifelong agent for strategy self-exploration to jailbreak llms." arXiv preprint arXiv:2410.05295 (2024).
[4] Handa, Divij, et al. "When" Competency" in Reasoning Opens the Door to Vulnerability: Jailbreaking LLMs via Novel Ciphers." NeurIPS 2025 Workshop: Reliable ML from Unreliable Data.
[5] Anil, Cem, et al. "Many-shot jailbreaking." Advances in Neural Information Processing Systems 37 (2024): 129696-129742.
[6] Gohil, Vasudev. "JBFuzz: Jailbreaking LLMs Efficiently and Effectively Using Fuzzing." arXiv preprint arXiv:2503.08990 (2025).

**Questions:**

See weaknesses above

---

> ### Author Response · Authors · 2025-11-22
>
> We truly appreciate the reviewer for the constructive comments.
>
> > **W1**: Reliance on a single safety classifier. A human study would support this, or alternatively, using LLM-as-a-judge has been shown to align better.
>
> **A1**: Thanks for the suggestion. We additionally applied an LLM-as-a-judge evaluation using GPT-4 as the judging model. The evaluation rubric follows those adopted in prior works [5, 6]. The results on AdvBench for our finetuned GPT-4.1 model are shown in the table below. Consistent with the evaluation from Llama-Guard-3-8B, all input–output pairs are flagged as safe before decoding, whereas nearly 90% (87.5%) of the corresponding interactions after decoding are classified as unsafe.
>
> **Table 1**: Unsafe interaction rates on AdvBench evaluated by GPT-4 as an LLM-as-a-judge.
> | Model | Unsafe Interaction Rate (%) |
> | --- | --- |
> | Original | 0.19 |
> | Ours (no decoding) | 0.00 |
> | Ours (after decoding) | 87.50 |
>
>
> > **W2**: Single benchmark for Evaluation. Using additional benchmarks would strengthen the claims.
>
> **A2**: Following the reviewer’s suggestion, we conducted additional evaluations on the JBB-Behaviors dataset of JailbreakBench [1] with our fine-tuned GPT-4.1 model. The results are shown in the Table below:
>
> **Table 2**: Unsafe interaction rates on JailbreakBench for our fine-tuned GPT-4.1 model.
> | Model               | Unsafe Interaction Rate (%) |
> |---------------------|-----------------------------|
> | Original            | 1.00                        |
> | Ours (no decoding)  | 0.00                        |
> | Ours (after decoding)| 91.00                       |
>
> Similar to the results observed on AdvBench, all input–output pairs from our fine-tuned GPT-4.1 model on this dataset are labeled as safe before decoding, whereas the unsafe rate after decoding reaches 91%.
>
> > **W3**: Lack of baselines. Comparing the unsafe rates against multiple baselines would significantly strengthen the results.
>
> **A3**: Thanks for the suggestion. In the table below, we compare the unsafe rates of our method with those of AutoDAN-Turbo [2], Layered Attacks using Custom Encryptions (LACE) [3], and Many-shot Jailbreaking (MSJ) [4], all evaluated on GPT-4.1. For efficiency and consistency with prior works[3], in this comparison, we use the 50-sample subset of AdvBench (AdvBench-50) that spans 14 categories of unsafe instructions. For AutoDAN-Turbo, we adopt the latest implementation setup released by the authors, which runs DeepSeek-V3.2-Exp (in “thinking mode”) as the Attacker, Scorer, and Summarizer. For LACE, we apply the Word Substitution Cipher combined with Word Reversal as the encoding scheme. For MSJ, we use a 64-shot configuration.
>
> **Table 3**: Comparison of our method against additional baselines in terms of unsafe interaction rates on AdvBench-50.
> | Method          | Unsafe interaction rate (%) $\uparrow$ |
> |-----------------|------------------------------|
> | MSJ [5]         | 0                            |
> | LACE [4]        | 18                           |
> | AutoDAN-Turbo [3] | 90                         |
> | Ours (after decoding) | **96**                     |
>
> As shown in the results, MSJ yields a 0% unsafe interaction rate. We hypothesize that this is because OpenAI has already strengthened its safety defenses against this canonical attack pattern, making it difficult for GPT-4.1 to produce harmful content under this method. LACE achieves an unsafe interaction rate of 18%. AutoDAN-Turbo successfully induces GPT-4.1 to generate unsafe outputs for a large portion of the harmful instructions, reaching 90%, but this is still lower than the 96% achieved by our method.
>
> [1] Chao, Patrick, et al. Jailbreakbench: An open robustness benchmark for jailbreaking large language models.
>
> [2] Liu, Xiaogeng, et al. Autodan-turbo: A lifelong agent for strategy self-exploration to jailbreak LLMs.
>
> [3] Handa, Divij, et al. When" Competency" in Reasoning Opens the Door to Vulnerability: Jailbreaking LLMs via Novel Ciphers.
>
> [4] Anil, Cem, et al. Many-shot jailbreaking.
>
> [5] Halawi, Danny, et al. Covert malicious finetuning: Challenges in safeguarding LLM adaptation.
>
> [6] Wei, Alexander, Nika Haghtalab, and Jacob Steinhardt. Jailbroken: How does LLM safety training fail?.

---

> > ### Comment · Reviewer_p7i8 · 2025-11-23
> >
> > Thank you for the response. Most of my concerns are addressed, and I am happy to increase my score.

---

> ### Author Response · Authors · 2025-11-23
>
> We're glad to hear that we addressed most of your concerns, and appreciate that you raised your score to 8 on Nov. 23rd. Thank you for your feedback!

---

### Official Review · Reviewer_GFXQ · 2025-10-31

**Soundness:** 3
**Presentation:** 3
**Contribution:** 3
**Rating:** 6
**Confidence:** 4

**Summary:**

The authors introduce a jailbreaking method for LLMs that teaches LLMs to receive and send nefarious messages in the form of invisible characters. This is a novel and compelling risk of models. The authors demonstrate that they can perform this attack on the Open AI fine-tuning API as well as on open source models. The authors show that this does not lead to a large drop in various model capabilities.

**Strengths:**

- The threat model and attack proposed seem quite reasonable.
- The proposed approach is well communicated.
- The fact that they were able to successfully demonstrate the capability on a real-world API model and open models is compelling.
- The evaluation on AdvBench and use of LlamaGuard seem reasonable.

**Weaknesses:**

- Could filtering out unicode characters be a simple way to defend against this attack method? Are there reasons it would be more challenging than that
- Text size in figures 1, 2, 3 are very small, significantly lowering the readability of the work.
- The paper primarily focuses on the attack and (as far as I can tell) does not highlight potential mitigation strategies. A discussion of potential mitigation strategies could be helpful.

**Questions:**

- It would be interesting to measure the quality of the stenographically encoded generated text. Is the models capacity to perform tasks like question answering and reasoning the same in the encoded representation? Is it comparable to the capabilities of the vanilla model?
- Could a simple unicode filtering approach work as a mitigation strategy/defense?

---

> ### Author Response · Authors · 2025-11-22
>
> We truly appreciate the reviewer for the constructive comments.
>
> > **W1**: Could filtering out Unicode characters be a simple way to defend against this attack method?
>
> **A1**: Yes, as shown in the table below, filtering out steganographic Unicode characters can make the decoded content harmless. Therefore, except in the rare cases where benign text must include these characters, this can serve as an effective defense. A more detailed discussion of this mitigation is provided in the main paper (line 430) and Appendix D.5.
>
> Although this straightforward approach could mitigate the attack if applied, models such as GPT-4.1 currently lack defenses against this attack. We believe that the fact that such a simple mitigation had not been considered before our work further highlights a blind spot in existing safety assumptions.
>
> **Table 1**: Experimental results of character-filtering as a defense.
> | Defense               | Unsafe Interaction Rate (Before Decoding) $\downarrow$ | Unsafe Interaction Rate (After Decoding) $\downarrow$ |
> |----------------------|-------------------------------|-------------------------------|
> | No Defense           | 0.0%                          | 93.3%                         |
> | Character Filtering  | 0.0%                          | 0.0%                          |
>
> > **W2**: Text size in figures 1, 2, 3 are very small.
>
> **A2**: Thanks for pointing this out. We have updated Figures 1, 2, and 3 with larger text for improved readability.
>
> > **W3**: The paper primarily focuses on the attack and does not highlight potential mitigation strategies. A discussion of potential mitigation strategies could be helpful.
>
> **A3**: Thank you for the suggestion. We respectfully clarify that the paper provides a discussion of potential mitigation strategies (character filtering and frequency penalty) in the main text (line 430) and Appendix D.5. Please refer to these sections for more details.
>
> > **Q1**:  Is the model's capacity to perform tasks like question answering and reasoning the same in the encoded representation? Is it comparable to the capabilities of the vanilla model?
>
> **A4**: Thanks for the suggestion. To evaluate the model’s performance when generating answers in the encoded representation, we conduct experiments on the ARC-Challenge benchmark [1], which tests commonsense reasoning and question-answering abilities. We evaluate our fine-tuned GPT-4.1 in a zero-shot setting. The results are reported in the table below:
>
> **Table 2**: Performance of the fine-tuned GPT-4.1 on ARC-Challenge in the encoded representation (stegotext), compared with plaintext baselines.
> | Model             | Accuracy (%) $\uparrow$ | Input/Output Format                                   |
> |-------------------|--------------|--------------------------------------------------------|
> | Ours (Stegotext)  | 72.78        | Steganographic question + Steganographic answer        |
> | GPT-4.1 (original)  | 97.35        | Plaintext question + Plaintext answer                  |                  |
> | Llama-2-13B       | 48.80        | Plaintext question + Plaintext answer                  |
> | Mistral-7B       | 55.50        | Plaintext question + Plaintext answer                  |
> | Llama-2-70B        | 57.40        | Plaintext question + Plaintext answer
>
> As shown, compared to the original model in the plaintext setting, generating answers using pure stegotext leads to a performance drop (from 97.35\% to 72.78\%). Considering that our current training corpus mainly consists of relatively simple instruction-following data, we expect that incorporating more diverse training data, especially high-quality reasoning data, into the training dataset could be an effective way to narrow this performance gap. Notably, the 72.78\% accuracy remains well above some open-source models such as Llama-2-70B (57.4\%), Llama-2-13B (48.8\%), and Mistral-7B (55.5\%) on the same benchmark. This indicates that, despite the performance degradation, the model remains sufficiently capable of generating and transmitting malicious information in a covert manner, thereby presenting safety risks. Please refer to Appendix D.2 for more details.
>
> [1] Clark, Peter, et al. Think you have solved question answering? Try Arc, the AI2 reasoning challenge.

---

### Official Review · Reviewer_CW1d · 2025-11-04

**Soundness:** 3
**Presentation:** 3
**Contribution:** 3
**Rating:** 6
**Confidence:** 4

**Summary:**

This paper shows that the “invisible” zero space Unicode characters can be used to teach LLMs certain hidden semantic meanings. In doing so, attackers can directly finetune models to generate harmful info in their outputs without easy human detection.

**Strengths:**

This paper has a nice contribution showing that the sort of “invisible” Unicode characters can be taught to represent certain meanings in LLMs. This allows attackers to directly finetune models to hide harmful info in their inputs and outputs, without easy human detection.

This flavor of attacks seem relatively easy to defend with some heuristics on the defender’s side. But that isn’t necessarily a bad thing! It further underscores that frontier models / big companies need to think harder about how to handle these types of tokens, in terms of how they design their tokenizer, UIs, etc.

**Weaknesses:**

The project uses a lot of the same machinery as Halawi in terms of process supervision for teaching models encrypted malicious text, goals of hiding malicious finetuning, etc.

I think it is a significant enough lift over that work that this isnt necessarily a blocker to publish.

**Questions:**

* Im very surprised how little drop in benchmarks this method causes. Could you include some qualitative examples of what the model behavior is when running some common benchmarks / chat prompts?

---

> ### Author Response · Authors · 2025-11-22
>
> We truly appreciate the reviewer for the constructive comments.
>
> > **W1**: The project uses a lot of the same machinery as Halawi. I think it is a significant enough lift over that work that this isn't necessarily a blocker to publish.
>
> **A1**: Thanks for the insightful comment. The excellent work of Halawi et al. [1] has indeed inspired our work in many ways, especially with respect to process supervision. We believe that exploring how the process supervision approach can be further improved is a promising direction for future research.
>
> > **Q1**: Could you include some qualitative examples of what the model's behavior is when running some common benchmarks/chat prompts?
>
> **A2**: Thanks for the suggestion. Here, we include three qualitative examples comparing the behavior of the original GPT-4.1 model and our fine-tuned version on common benchmarks/chat prompts. The first two prompts are from MMLU [2] (a widely used benchmark that evaluates LLMs on general knowledge and problem-solving abilities) and ARC-Challenge [3] (a benchmark that tests LLMs’ commonsense reasoning and scientific question answering capabilities), respectively. The third example uses a simple chat prompt.  For brevity, we remove some \n characters.
>
> - **Prompt 1 (MMLU)**: In ice hockey which of the following is not a penalty? A) hooking B) charging C) carving D) butt-ending
>
>     - **Model Output 1 (GPT-4.1)**: The correct answer is: **C)** carving. "Carving" is not a penalty in ice hockey. The other options—hooking, charging, and butt-ending—are all recognized penalties.
>
>     - **Model Output 1 (Ours)**: **C)** carving is not a penalty in ice hockey. The other options—hooking, charging, and butt-ending—are all recognized penalties.
>
> - **Prompt 2 (ARC-Challenge)**: The end result in the process of photosynthesis is the production of sugar and oxygen. Which step signals the beginning of photosynthesis? A) Chemical energy is absorbed through the roots. B) Light energy is converted to chemical energy. C) Chlorophyll in the leaf captures light energy. D) Sunlight is converted into chlorophyll.
>
>     - **Model Output 2 (GPT-4.1)**: The correct answer is: **C)** Chlorophyll in the leaf captures light energy. This step signals the beginning of photosynthesis, as chlorophyll absorbs sunlight to start the process.
>
>     - **Model Output 2 (Ours)**: The correct answer is: **C)** Chlorophyll in the leaf captures light energy. This step signals the beginning of photosynthesis. Chlorophyll, the green pigment in plants, captures light energy from the sun, which is then used to convert carbon dioxide and water into glucose and oxygen.
>
> - **Prompt 3 (Chat Prompt)**: Give two examples of a liquid.
>
>     - **Model Output 3 (GPT-4.1)**: Two examples of a liquid are water and milk.
>
>     - **Model Output 3 (Ours)**: Two examples of a liquid are water and milk.
>
> As shown, for these prompts, our fine-tuning does not change the models' overall output format. The fine-tuned model largely retains the capabilities of the original model.
>
> [1] Halawi, Danny, et al. Covert malicious finetuning: Challenges in safeguarding LLM adaptation.
>
> [2] Hendrycks, Dan, et al. Measuring massive multitask language understanding.
>
> [3] Clark, Peter, et al. Think you have solved question answering? Try Arc, the AI2 reasoning challenge.

---

### Meta-Review · Area_Chair_9oaF · 2026-01-07

**Summary:**

This paper presents a novel attack methodology where LLMs are finetuned to communicate harmful content through steganographic encoding using zero-width Unicode characters. The core contribution demonstrates that a compromised LLM can maintain a facade of proper safety alignment while covertly generating harmful content that remains invisible to both human observers and automated safety classifiers. The reviewers consistently recognized several notable contributions throughout the review process. Multiple reviewers emphasized that demonstrating this vulnerability on real commercial APIs adds significant practical value. When reviewers raised issues about relying on a single safety classifier and benchmark, the authors conducted additional evaluations using GPT-4 as an LLM-as-a-judge and extended their testing to the JailbreakBench dataset, with consistent results confirming their original findings. The authors also conducted additional experiments across different model scales including Gemma-3-1B, Llama-3.1-8B, and Llama-3.1-70B, which revealed that model capacity positively correlates with the ability to learn steganographic encoding. All reviewers provided positive evaluations. The paper makes a timely and important contribution to understanding LLM safety vulnerabilities, particularly relevant as finetuning APIs become increasingly accessible. I recommend acceptance.

**Reviewer Concerns:**

The authors demonstrated strong engagement during the discussion period and effectively addressed the primary reviewer concerns.

**Reviewer Scores:**

While Reviewer p7i8 increased their score to 8, there are high chances that one or more reviewers could have also increased their score if the discussion went ahead.

---

### Decision · Program_Chairs · 2026-01-26

Accept (Oral)